# Effect of Operational Parameters on the Removal of Carbamazepine and Nutrients in a Submerged Ceramic Membrane Bioreactor

**DOI:** 10.3390/membranes12040420

**Published:** 2022-04-14

**Authors:** Khanh-Chau Dao, Chih-Chi Yang, Ku-Fan Chen, Yung-Pin Tsai

**Affiliations:** 1Department of Civil Engineering, National Chi Nan University, Nantou Hsien 54561, Taiwan; daokhanhchau07@gmail.com (K.-C.D.); chi813@gmail.com (C.-C.Y.); kfchen@ncnu.edu.tw (K.-F.C.); 2Department of Health, Dong Nai Technology University, Bien Hoa 810000, Dong Nai, Vietnam

**Keywords:** full-factorial design, carbamazepine, membrane bioreactor, hospital wastewater, operating parameters

## Abstract

Pharmaceuticals and personal care products have raised significant concerns because of their extensive use, presence in aquatic environments, and potential impacts on wildlife and humans. Carbamazepine was the most frequently detected pharmaceutical residue among pharmaceuticals and personal care products. Nevertheless, the low removal efficiency of carbamazepine by conventional wastewater treatment plants was due to resistance to biodegradation at low concentrations. A membrane bioreactor (MBR) has recently attracted attention as a new separation process for wastewater treatment in cities and industries because of its effectiveness in separating pollutants and its tolerance to high or shock loadings. In the current research, the main and interaction effects of three operating parameters, including hydraulic retention time (12–24 h), dissolved oxygen (1.5–5.5 mg/L), and sludge retention time (5–15 days), on removing carbamazepine, chemical oxygen demand, ammonia nitrogen, and phosphorus using ceramic membranes was investigated by applying a two-level full-factorial design analysis. Optimum dissolved oxygen, hydraulic retention time, and sludge retention time were 1.7 mg/L, 24 h, and 5 days, respectively. The research results showed the applicability of the MBR to wastewater treatment with a high carbamazepine loading rate and the removal of nutrients.

## 1. Introduction

In recent years, pharmaceuticals and personal care products (PPCPs) have caused growing concerns as emerging contaminants because of their extensive use, presence in aquatic environments, and potential impacts on wildlife and humans. PPCPs comprise a large and varied group of organic compounds, including pharmaceutical drugs and components of daily personal care products (soaps, lotions, toothpaste, fragrances, sunscreens, etc.) as well as their metabolites and transformation products that are widely used in large quantities around the world [1]. Previous studies showed that many PPCPs were frequently detected in surface water, groundwater, seawater, and drinking water [2,3,4].

Pharmaceuticals and other drug components are widely used in hospitals, so hospital effluents generally have higher detection rates and concentrations of these contaminants [5,6]. One of the main environmental problems caused by hospital effluents is their discharge into urban sewerage systems, without preliminary treatment, and entering into the water bodies. Therefore, treatment plants should be upgraded to eliminate these PPCPs to the greatest possible extent before their effluents are released into the environment. In recent investigations, different antibiotics were found in low concentrations in municipal wastewater effluents and surface water [7,8,9]. Antibiotics in water are an environmental concern because they could disturb microbial ecology, increase the proliferation of antibiotic-resistant pathogens, and threaten human health.

Carbamazepine (CBZ) is a common drug that controls seizures, with about 1014 tons of it consumed worldwide annually. Overdosing on CBZ and its metabolites, on the other hand, can harm the human liver and emopoietic systems. As a result, it was the most common pharmaceutical residue found in water bodies [10,11,12]. Investigations found that CBZ was persistent with removal efficiency of the WWTPs being mostly below 10% [13], due to its resistance to biodegradation at low concentrations and its less attachment to sludge [14]. Researchers considered that CBZ could be a “witness molecule” confirming the presence and persistence of drugs in water bodies [15].

In recent years, membrane technology has attracted attention as a new separation process for water and wastewater treatment in cities and industries. Combining membranes with biological treatments is an attractive technique and has resulted in a new concept: a membrane bioreactor (MBR). An MBR was first used to treat wastewater over 50 years ago [16]. Submerged MBRs (SMBR), characterized by immersing the membrane modules as separation units directly in the bioreactor, were developed for wastewater treatment in the 1990s [17]. Membrane filtration processes are promising alternatives for eliminating PPCPs from wastewater [18,19]. Although their effectiveness in the separation of pollutants, tolerance to high or shock loadings, stable and excellent effluent quality, ease of operation, small footprint, and effective bacterial elimination, they are currently facing some research and developmental challenges, such as membrane fouling, high membrane cost, and the need for pre-treatment [20].

The current research investigated operating parameters (factors) via FFD experiments to optimize the MBR process. First, the important operating conditions, such as hydraulic retention time—RT, dissolved oxygen—DO, and sludge retention time—SRT, were chosen to study their main and interaction effects on the following responses: CBZ, chemical oxygen demand—COD, ammonia nitrogen—NH_4_^+^-N, and phosphorus—PO_4_^3−^-P removal. In addition, the ranges of the factors were selected based on the capability of the experimental setup, economic considerations, and membrane operating limits. Finally, a regression model was presented for each response, and optimization of the process was carried out to maximize the removal of CBZ, COD, NH_4_^+^-N, and PO_4_^3−^-P.

## 2. Materials and Methods

### 2.1. Chemicals

Chemicals used to prepare synthetic wastewater were analytical grade, acetonitrile (HPLC grade), acetone, ethyl acetate, and methanol (HPLC grade) purchased from Sigma-Aldrich, St. Louis, MO, USA. An Oasis HLB 3 cc Vac Cartridge was supplied by Waters, Milford, MA, USA.

CBZ was of the highest purity commercially available, and it was purchased from Sigma-Aldrich. Ultra-pure water was prepared with a Milli-Q water purification system (resistivity of 18.2 MΩ cm at 25 °C). All solvents for HPLC application were filtered before use with the 0.45-μm membrane filter paper (Millipore, Merck, Darmstadt, Germany) and degassed by ultrasonication for 30 min before use.

Stock solutions of CBZ were prepared in Milli-Q water from powdered substances at 1 g/L. The stock solution was made weekly from powdered substances and stored in the dark at 4 °C.

### 2.2. Simulated Wastewater

Synthetic hospital wastewater was composed of the following (g/L): peptone 0.12, meat extract 0.083, NH_4_Cl 0.143, NaCl 0.005, CaCl_2_·2H_2_O 0.003, MgSO_4_·7H_2_O 0.0015, CuCl_2_·2H_2_O 50 × 10^−6^, K_2_HPO_4_·3H_2_O 0.084, C_6_H_12_O_6_ 0.19, and NaHCO_3_ 0.83. This resulted in concentrations of COD 485.82 ± 57.11 mg/L, NH_4_^+^-N 60.72 ± 8.67 mg/L, PO_4_^3−^-P of 14.74 ± 1.09 mg/L, and a pH of 7.7 ± 0.2. CBZ was spiked into synthetic wastewater at 100 μg/L.

### 2.3. Bioreactor Configuration and Operation

The submerged MBR operated in a working volume of 25 L flat-sheet ceramic membrane module (GVE Environmental Co., Ltd., Taoyuan, Taiwan) with a nominal pore size of 0.1 μm and an effective area of 0.25 m^2^, Figure 1. The Al_2_O_3_-based flat-sheet ceramic membrane had three layers: a surface layer, a transition layer, and a support layer, as shown in the scanning electron microscopy (SEM, see Appendix A) image. The membrane was operated under an on–off mode (8 min on and 2 min off cycle) to relax the membrane module. The influent flow rate was adjusted to equal the effluent flow rate to maintain a constant water level. The systems were controlled automatically by timers and a pressure gauge. Air diffusers were positioned at the bottom of the reactor and the rear end of the membrane module for aeration and air scouring while the air supply was controlled by an airflow meter.

The transmembrane pressure (TMP) was measured using a pressure gauge installed between the membrane module and the permeate pump. The pressure gauge recorded the TMP daily. At the end of each experiment, membrane cleaning was performed. The membrane module was flushed with tap water to remove the visible cake layer and then immersed for a minimum of 24 h in a sodium hypochlorite solution of 3 ‰ (*v*/*v*). Activated sludge was removed from the reactor during chemical cleaning operations.

The seed-activated sludge was collected from a conventional wastewater system at National Chi Nan University. The ratio of the mixed liquor volatile suspended solids to mixed liquor suspended solids (MLVSS/MLSS) of the seed-activated sludge was 0.8. Activated sludge was maintained in a batch reactor at a DO of 2 mg/L, HRT of 48 h, and SRT of 20 days. The components in the synthetic wastewater were adjusted to maintain a BOD: N: P ratio of 100:15:5. In addition, nitrogen and phosphorus were put in as excess into the synthetic wastewater, so it was not deficient in essential nutrients for bacterial activity. After each experiment, an amount of sludge was added to the MBR reactor to reach a concentration of 5000 mg/L and acclimatized for 3 days before the experiment.

### 2.4. Modeling by Full-Factorial Design (FFD)

In this work, the FFD was employed to identify the crucial factors, the possibility of estimating interactions, and optimizing the parameters. The HRT (12 and 24 h), DO (1.5 and 5.5 mg/L), and SRT (5 and 15 days) were set, as in Table 1. Experiments were carried out at ambient temperature. CBZ, COD, NH_4_^+^-N, and PO_4_^3−^-P concentrations were analyzed. All samples were performed in triplicate, and the average standard deviation was calculated for each sample. Each experiment operated for five days.

The method consists of adding center points to the two-level FFD to protect curvature and allow an independent estimate of the error [21]. This method could also be easily upgraded to respond surface designs for further optimizations [22]. The regression equation based on the first-order model with three parameters and their interaction terms could be given in the form of the following expression [23]:Y_i_ = b_0_ + b_1_X_1i_ + b_2_X_2i_ + b_3_X_3i_ + b_12_X_1i_X_2i_ + b_13_X_1i_X_3i_ + b_23_X_2i_X_3i_ + b_123_X_1i_X_2i_X_3i_(1)
where Y_i_ is the response; X_ji_ values (j = 1, 2, 3; i = 1, 2, 3, …, 8) indicate the corresponding parameters in their coded forms; b_0_ is the average value of the result; b_1_, b_2_, and b_3_ are the linear coefficients; and b_12_, b_13_, b_23_, and b_123_ represent the interaction coefficients [23]. Adding interaction terms to the main effects introduced curvature into the response function. Therefore, if there was slight curvature in a limited region, a first-order model with interactions was appropriate for modeling [24,25]. Design-Expert^®^ 11 was utilized to design the experiments, and analysis of variance (ANOVA) was used to analyze the results.

### 2.5. Analytical Methods

Standard analytical methods [26] were applied in determining COD, MLSS, and PO_4_^3−^-P. COD was measured by the colorimetric method in the presence of potassium dichromate, and the absorbance was measured at 600 nm using a UV spectrometer (DR 5000, Hach, CO, USA). NH_4_^+^-N was measured by the indophenol method [27].

Before extraction, suspended solids in the samples were removed by filtering the samples through a 0.45-μm glass-fiber filter (Millipore, Merck, Darmstadt, Germany). Next, CBZ was extracted from the water samples using a selected cartridge. Before loading the sample, the solid-phase adsorbent was preconditioned with 5 mL of methanol followed by 5 mL of Milli-Q water. The sample was then passed through the cartridge at a 5 mL/min flow rate. Subsequently, the cartridge was eluted with five 1 mL aliquots of ethyl acetate–acetone (50:50, *v*/*v*) at a rate of 1 mL/min; the combined aliquots were evaporated under a gentle flow of high purity nitrogen and redissolved in 1 mL of methanol. The analyses of CBZ were carried out on an Agilent 1200 HPLC equipped with a G1329 autosampler, a G1315D diode array detector, and a G1316A column oven (Agilent Technologies Co. Ltd., Santa Clara, CA, USA). The detection wavelength was 210 nm, and the column temperature was set at 30 °C. An Eclipse XDB-C18 column (4.6 × 150 mm, particle size five μm, Agilent) was used for separation. The mobile phase was acetonitrile–water (31:69, *v*/*v*) at a 1 mL/min flow rate. The injection volume was 20 μL.

## 3. Results and Discussion

### 3.1. Biodegradation Efficiency and CBZ Removal

Table 2 gives an overview of the results achieved by the MBR treatment. The system showed a relatively low efficiency in removing CBZ with an average removal of 18.41% due to its recalcitrance, but higher than previous studies by MBR using synthetic wastewater, which was commonly below 13% [28,29,30].

However, the study results also demonstrated the effectiveness of the MBR system in removing COD and NH_4_^+^-N, with an average removal efficiency of 86.45 and 90.55%, respectively. On the other hand, the experimental results showed that the MBR system could not remove phosphorus effectively when the outlet concentration was higher than the inlet.

### 3.2. Model Fitting and Statistical Analysis

A total of 11 experiments were performed using a three-factor two-level FFD with three replicates at the center point. Table 3 shows the experimental design matrix, and the results of the response variables studied.

It was suggested that for a good fit of a model, R^2^ should be at least 0.80. These response variables had an R^2^ greater than 0.80, indicating that the regression models explained the reaction well [31]. The experiments were carried out in randomized runs to determine the effect of the factors on four characteristic responses: CBZ, COD, NH_4_^+^-N, and PO_4_^3−^-P.

Percent contributions of all factors are presented as a chart in Figure 2 to determine the importance of factors. It could be seen from Figure 2a,b, and d that in the case of CBZ, COD, and phosphorus removal, DO was the most influential factor. However, according to Figure 2c, HRT had the most significant effect on ammonia removal.

### 3.3. CBZ Removal

The overall performance of the MBR was estimated by calculating CBZ removal as a response. The three-factor interaction (3FI) model described the variation of the CBZ removal efficiency as a function of the variables. Analysis of variance (ANOVA) for the model terms is summarized in Table 4. The F-value and *p*-value determined the significance of each coefficient. It was observed from ANOVA analysis that the confidence level was greater than 80% (*p* < 0.05) for the CBZ removal response, while the F-value and *p*-value of the model were 38.44 and 0.0062, respectively. This indicated that the estimated model fitted the experimental data adequately.

Furthermore, the coefficient of determination R^2^ of the model was reasonably close to 1 (0.9890), implying that the model explained about 98.90% of the variability in the data. From Table 4, A (DO), C (SRT), and B (HRT) were significant model terms. The interaction between DO and SRT was more important than other interactions (AB, BC, and ABC), with a probability value larger than 0.05. After elimination of insignificant parameters, the final empirical model at 95% confidence level could be represented as:CBZ removal (%) = 10.16 + 0.54 × A + 2.01 × B − 0.15 × C − 0.33 × A × B − 0.097 × A × C − 0.098 × B × C + 0.02 × A × B × C(2)

Figure 3a–c shows the 3D surface plot of CBZ removal versus two varying parameters at a fixed value of the third parameter. Again, a decrease in CBZ removal efficiency was observed with an increase in DO and SRT, with the effect of DO being a little greater than that of SRT.

The figures show a slight positive effect of HRT on CBZ removal in the MBR system. The maximum CBZ removal efficiency was 38.36 ± 4.49% at 1.5 mg/L DO, 24 h HRT, and 5 days SRT, while the minimum reached 9.04 ± 1.12% at 5.5 mg/L DO, 12 h HRT, and 15 days SRT. This finding was similar to previous studies, which found that anoxic conditions removed more CBZ than aerobic conditions [32]. Its degradation was highly dependent on the operating conditions [33], the presence of strong electron-withdrawing groups, or the absence of electron-donating groups in CBZ, might explain the low removal effectiveness by activated sludge or MBR processes [34], even under long SRT [35]. This study suggested that a short SRT might have a significant impact on the research model, and the optimization of the operating factors could significantly improve the overall removal of CBZ.

The main mechanisms for removing PPCPs by MBR are biodegradation and sorption [36]. To eliminate CBZ, however, combining the MBR process with other treatments, such as advanced oxidation processes (AOPs) or adsorption, is required to lower the concentration of CBZ in the permeate.

### 3.4. COD Removal

The 3FI model describes the variation of the COD removal in the system studied. Based on ANOVA (Table 5), A (DO) and B (HRT) were significant model terms, which might be because of the increase in the aerobic heterotrophic bacteria, while the effect of C (SRT) was not much on the overall removal efficiency. Furthermore, the confidence level of ANOVA of the COD removal response was greater than 80% (*p* < 0.05) for COD response, while the F-value and *p*-value of the model were 19.38 and 0.0167, respectively. This also indicated that the estimated model fitted the experimental data adequately. It was further shown that the interactions of AB, AC, BC, and ABC were not significant model terms (factors).

Figure 4a–c illustrates the interactive effect of the variables on COD removal. The results showed changes in DO from 1.5 to 5.5 mg/L, and HRT from 12 to 24 h increased COD removal by about 13% and 11%, respectively, whereas the COD removal was lowered 5% for the changes in SRT. Overall, the system showed good performance for COD removal, with removal efficiencies ranging between 70% and 99%. COD removal efficiencies being high throughout the experiments could be due to the filtration membrane′s ability to retain all the particulate COD [37]. The maximum values for the response were 99.37 ± 0.42% at 5.5 mg/L DO, 24 h HRT, and 5 days SRT compared to the minimum of 69.23 ± 2.56% at 1.5 mg/L DO, 12 h HRT, and 15 days SRT. Therefore, the final empirical model at 95% confidence level could be represented as:COD removal (%) = 71.43 + 1.93 × A + 0.60 × B − 1.45 × C + 0.04 × A × B + 0.20 × A × C + 0.04 × B × C − 0.007 × A × B × C(3)

Positive coefficients indicated an increasing effect of A and B on the response, +1.93 and +0.60, respectively, suggesting that the response was more dependent on DO (A) than HRT (B). The current study confirmed a small effect of SRT on COD removal efficiency at short SRT (3, 5, and 10 days) before [38]. This phenomenon could be due to a decrease in the ratio of the active biomass to that of the total biomass (MLVSS/MLSS) following increasing SRT, indicating that the increased sludge age could decrease microbial activities.

### 3.5. Ammonia Removal

Based on ANOVA (Table 6), A and B were significant model terms. In addition, the confidence level for the ammonia removal response was greater than 80% (*p* < 0.05), while the model′s F-value and *p*-value were 20.57 and 0.0154, respectively. This suggested that the estimated model adequately matched the experimental data. Furthermore, the model’s coefficient of determination R^2^ was relatively close to 1 (0.9796), meaning that the model described around 97.96% of the variability in the data.

Positive coefficients indicated an increasing effect of A and B on the response, +1.39 and +0.99, respectively. As could be seen, the effect of SRT on the response was lower than that of DO and HRT, while the interactions of AB, AC, BC, and ABC were not significant model terms. As a result, the maximum ammonia removal efficiency was 99.71% at DO (5.5 mg/L), HRT (24 h), and SRT (15 days). The final empirical model at 95% confidence level could be represented as:Ammonia removal (%) = 61.29 + 1.39×A + 0.99 × B + 1.72 × C + 0.05 × A × B − 0.18 × A × C − 0.07 × B × C + 0.007 × A × B × C(4)

The variation of ammonia removal as a function of the variables is shown in Figure 5a–c. It was observed that an increase in ammonia removal was due to increased DO, HRT, and SRT. According to several studies on nitrification in MBRs, increasing SRT improved ammonia removal efficiency significantly [39], while others showed that the high removal efficiency of ammonia was almost independent of SRT [40]. Membrane filtration increased the system′s performance by retaining all suspended solids, proteins, and polysaccharides from the sludge supernatant.

### 3.6. Phosphorus Removal

The biological phosphorous removal process is divided into anaerobic and aerobic stages. In the anaerobic zone, phosphate accumulating organisms (PAOs) release phosphorus and accumulate poly hydroxybutyrate (PHB), whereas, in the aerobic zone, phosphorous is absorbed [41].

According to ANOVA, the confidence level for phosphorus removal response was greater than 80% (*p* < 0.05), while the model′s F-value and *p*-value were 12.77 and 0.0303, respectively. This indicated that the estimated model fitted the experimental data well. Furthermore, the model′s coefficient of determination R^2^ was quite close to 1 (0.9675), indicating that the model described roughly 96.75% of the data variability.

A and B were significant model terms, as shown in Table 7. Positive coefficients of +0.90, +0.17, and +0.35 indicated an increasing A, B, and C effect on the response, respectively. As could be observed, SRT had a lower effect on the response than DO and HRT, and the interactions between AB, AC, BC, and ABC were not significant model terms. Therefore, the final empirical model at 95% confidence level could be represented as follows:Phosphorus removal (%) = −7.28 + 0.90 × A + 0.17 × B + 0.35 × C − 0.12 × A × B − 0.11 × A × C − 0.02 × B × C + 0.005 × A × B × C(5)

Phosphorus removal efficiency is inversely proportional to DO, HRT and SRT. As a result, the maximum phosphorus removal achieved was −5.91% at low DO (1.5 mg/L), HRT (12 h), and high SRT (15 days), indicating an anaerobic–aerobic environment was provided at this condition. Figure 6a–c shows the interactive effects of the variables on phosphorus removal.

### 3.7. Process Optimization

Optimization of the operating parameters based on two-level FFD was carried out to improve the MBR process. In addition, a multi-response method called the desirability function was used, which found operating conditions that yielded the “most desirable” responses [42,43].

In this method, multiple responses could be combined into the “desirability function” by choosing a value from 0 to 1. This method transformed an adequate function of each determined response level (Yi) into a desirability score (di) within a 0–1 scale. After that, all individual desirability scores were integrated into a single overall desirability function optimized to determine the optimum set of input variables [42].

For this reason, operating parameters were set to be within the range, whereas CBZ, COD, ammonia, and phosphorus removal efficiency were set to maximum. Figure 7 shows the graphical desirability generated from 40 optimum points. At the best point with a maximum overall desirability of 0.72, the optimum DO, HRT, and SRT were found to be 1.7 mg/L, 24 h, and 5 days, respectively. Under optimum conditions, CBZ, COD, ammonia, and phosphorus removal were obtained at 37.08, 88.23, 90.12, and −7.48 %, respectively.

## 4. Conclusions

The current work used the FFD methodology to determine the significant parameters, investigate their interactions, and optimize conditions for the MBR process concerning CBZ, COD, ammonia, and phosphorus removal. As a result, the following conclusions were drawn:Significant analysis of main and interaction effects revealed that the relative importance of significant parameters and interaction factors could be observed as follows: (a) for CBZ removal, when DO (A) and SRT (C) were increased, a decrease in removal efficiency was observed, with DO’s effect being a little greater than that of SRT, while a short SRT might significantly impact the research model. HRT had a slight positive effect on CBZ removal; (b) COD removal: the response was more dependent on A than B while confirming a small effect of C on removal efficiency. The AB, AC, BC, and ABC interactions were not significant model terms. The system showed good performance for COD removal, with removal efficiencies ranging between 70% and 99% over the experiments; (c) for ammonia removal, positive coefficients indicated an increasing effect of A and B on the response, while the AB, AC, BC, and ABC interactions were not significant model terms. It was observed that an increase in removal rate was due to increased DO, HRT, and SRT; (d) for phosphorus removal, A and B were significant model terms. The interactions between AB, AC, BC, and ABC were not significant model terms, and removal efficiency was inversely proportional to DO, HRT, and SRT.Optimization of the process was found at DO, HRT, and SRT of 1.7 mg/L, 24 h, and 5 days for maximum CBZ, COD, ammonia, and phosphorus removal that obtained removal efficiencies 37.08, 88.23, 90.12, and −7.48 %, respectively.The flat-sheet ceramic MBR demonstrated efficiency removals as high as 38.36 ± 4.49%, as CBZ is known to be a somewhat recalcitrant compound.To eliminate CBZ in the permeate, future studies require the combination of the MBR process with other treatments, such as advanced oxidation processes (AOPs) or adsorption.

## Figures and Tables

**Figure 1 membranes-12-00420-f001:**
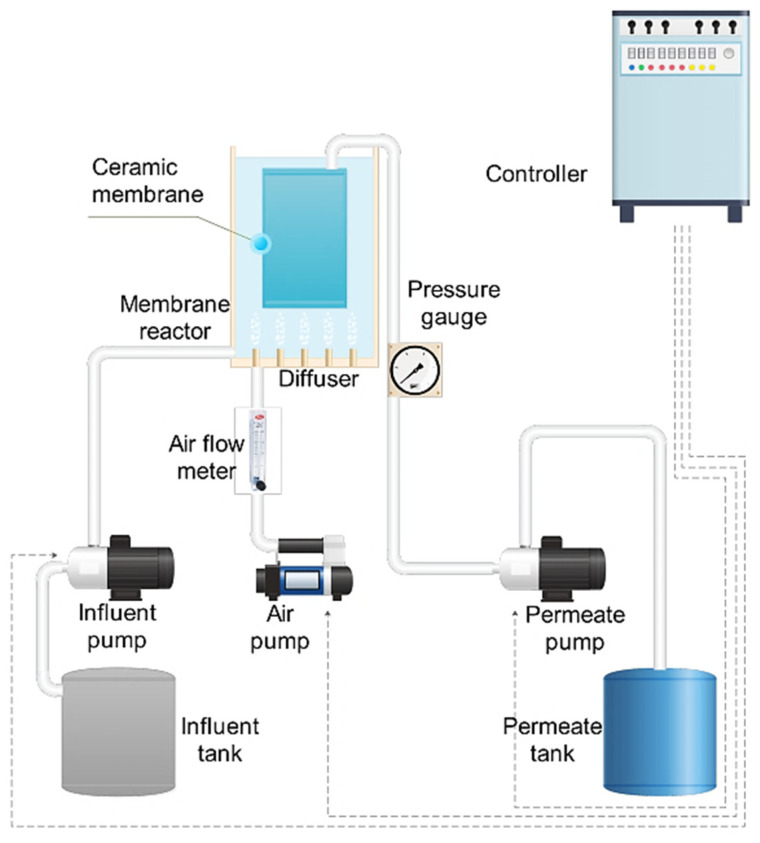
Schematic diagram of the Membrane bioreactor (MBR) experimental setup.

**Figure 2 membranes-12-00420-f002:**
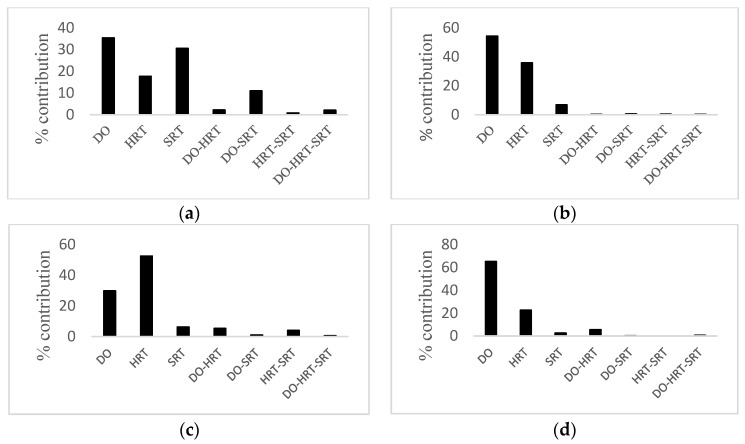
Percent contribution of each factor on the performance statistics of (**a**) CBZ removal, (**b**) COD removal, (**c**) ammonia removal, (**d**) phosphorus removal.

**Figure 3 membranes-12-00420-f003:**
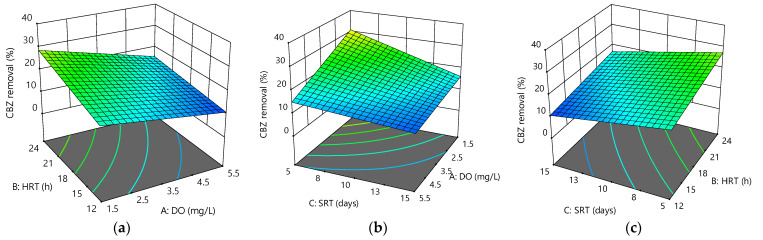
Response surface plots for CBZ removal efficiency as a function of the following: (**a**) HRT and DO at SRT = 10 days; (**b**) SRT and DO at HRT = 18 h; (**c**) HRT and SRT at DO = 3.5 mg/L.

**Figure 4 membranes-12-00420-f004:**
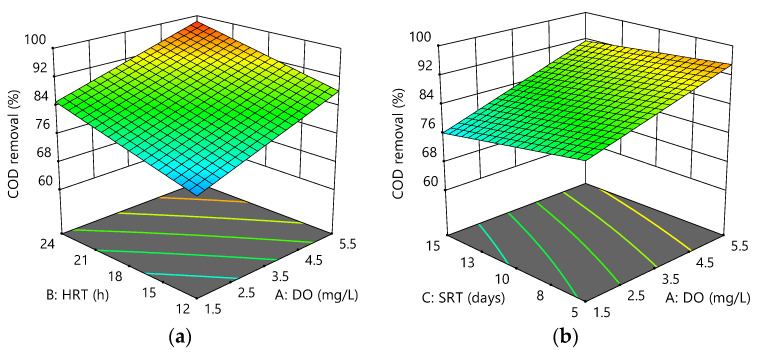
Response surface plots for COD removal efficiency as a function of the following: (**a**) HRT and DO at SRT = 10 days; (**b**) SRT and DO at HRT = 18 h; (**c**) HRT and SRT at DO = 3.5 mg/L.

**Figure 5 membranes-12-00420-f005:**
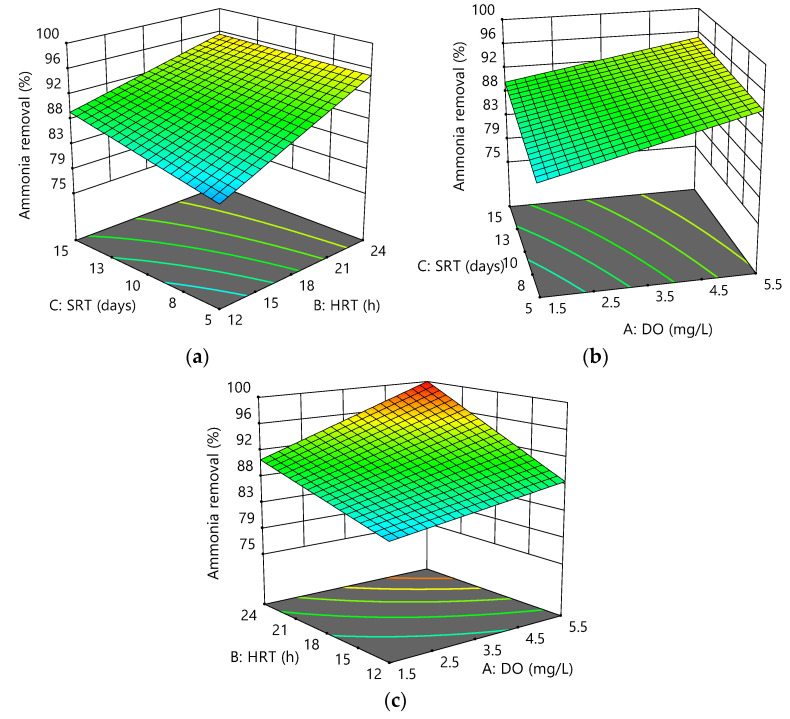
Response surface plots for ammonia removal efficiency as a function of the following: (**a**) HRT and DO at SRT = 10 days; (**b**) SRT and DO at HRT = 18 h; (**c**) HRT and SRT at DO = 3.5 mg/L.

**Figure 6 membranes-12-00420-f006:**
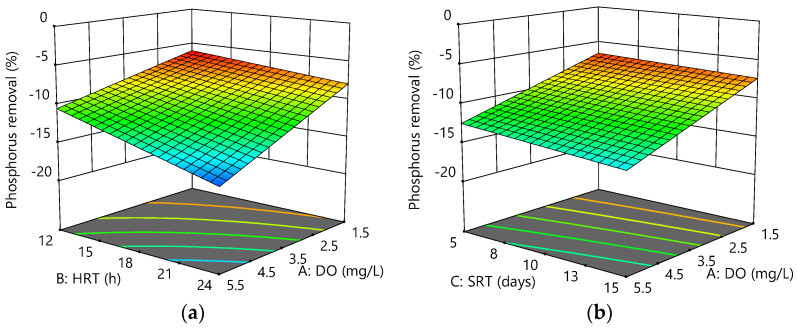
Response surface plots for phosphorus removal efficiency as a function of the following: (**a**) HRT and DO at SRT = 10 days; (**b**) SRT and DO at HRT = 18 h; (**c**) HRT and SRT at DO = 3.5 mg/L.

**Figure 7 membranes-12-00420-f007:**
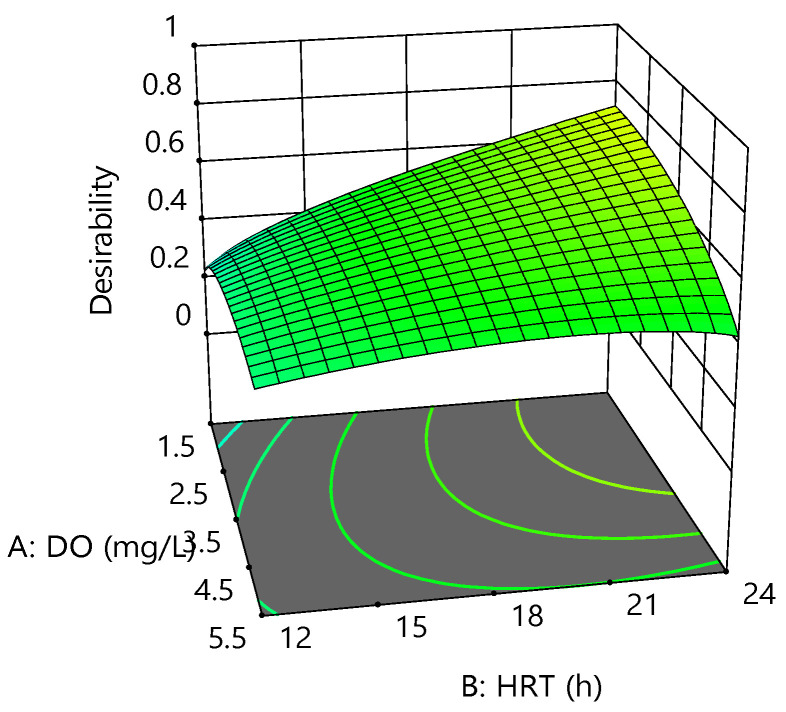
Desirability fitted 3D surface at an SRT of 5 days.

**Table 1 membranes-12-00420-t001:** Factors and levels for full-factorial design (FFD).

Factor	Name	Units	Type	Minimum	Maximum	Coded Low	Coded High
A	DO	mg/L	Numeric	1.5	5.5	−1 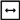 1.5	+1 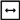 5.5
B	HRT	h	Numeric	12	24	−1 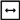 12	+1 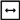 24
C	SRT	days	Numeric	5	15	−1 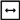 5	+1 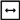 15

**Table 2 membranes-12-00420-t002:** CBZ, COD, ammonia, and phosphorus removal efficiency.

Removal (%)	Minimum	Maximum	Average
CBZ	9.04	38.36	18.42
COD	69.23	99.37	86.45
NH_4_^+^-N	79.75	99.71	90.55
PO_4_^3−^-P	−16.87	−5.91	−10.15

**Table 3 membranes-12-00420-t003:** Experimental design table for the factors and responses.

		Factor 1	Factor 2	Factor 3	Response 1	Response 2	Response 3	Response 4
Std	Run	A:DO	B:HRT	C:SRT	CBZ removal	COD removal	Ammonia removal	Phosphorus removal
		mg/L	h	days	%	%	%	%
9	1	3.5	18	10	17.66 ± 3.33	87.18 ± 3.88	91.62 ± 6.04	−9.57 ± 2.22
7	2	1.5	24	15	19.65 ± 7.18	82.16 ± 4.97	90.40 ± 1.11	−8.71 ± 1.05
4	3	5.5	24	5	17.23 ± 5.38	99.37 ± 0.42	99.58 ± 0.25	−15.43 ± 0.33
8	4	5.5	24	15	14.75 ± 4.48	95.97 ± 0.99	99.71 ± 0.03	−16.87 ± 1.51
6	5	5.5	12	15	9.04 ± 1.12	85.87 ± 2.37	89.38 ± 3.65	−11.51 ± 1.75
2	6	5.5	12	5	13.67 ± 4.81	89.46 ± 1.40	85.81 ± 1.38	−9.50 ± 2.95
10	7	3.5	18	10	16.20 ± 4.00	87.62 ± 3.23	90.35 ± 2.41	−11.84 ± 2.47
1	8	1.5	12	5	24.12 ± 1.45	77.49 ± 1.46	79.75 ± 6.58	−6.03 ± 1.73
11	9	3.5	18	10	18.48 ± 4.35	89.54 ± 0.22	92.62 ± 1.64	−9.20 ± 3.27
5	10	1.5	12	15	13.39 ± 10.31	69.23 ± 2.56	87.50 ± 3.51	−5.91 ± 1.23
3	11	1.5	24	5	38.36 ± 4.49	87.07 ± 0.54	89.30 ± 1.33	−7.10 ± 4.70

**Table 4 membranes-12-00420-t004:** ANOVA results for CBZ removal response.

Source	Sum of Squares	df	Mean Square	F-Value	*p*-Value	
Model	582.31	7	83.19	38.44	0.0062	Significant
A-DO	207.62	1	207.62	95.94	0.0023	
B-HRT	111.38	1	111.38	51.47	0.0056	
C-SRT	167.72	1	167.72	77.51	0.0031	
AB	16.02	1	16.02	7.40	0.0725	
AC	62.82	1	62.82	29.03	0.0125	
BC	4.13	1	4.13	1.91	0.2610	
ABC	12.62	1	12.62	5.83	0.0946	
Residual	6.49	3	2.16			
Lack of fit	3.81	1	3.81	2.85	0.2335	Not significant
Pure error	2.68	2	1.34			
Cor total	588.80	10				
Std. dev.	1.47		R^2^		0.9890	
Mean	18.41		Adjusted R^2^		0.9632	

**Table 5 membranes-12-00420-t005:** ANOVA results for COD removal response.

Source	Sum of Squares	df	Mean Square	F-Value	*p*-Value	
Model	659.56	7	94.22	19.38	0.0167	Significant
A-DO	374.40	1	374.40	77.02	0.0031	
B-HRT	226.00	1	226.00	46.49	0.0065	
C-SRT	50.80	1	50.80	10.45	0.0481	
AB	0.7858	1	0.7858	0.1617	0.7146	
AC	4.76	1	4.76	0.9788	0.3954	
BC	1.56	1	1.56	0.3218	0.6102	
ABC	1.25	1	1.25	0.2570	0.6471	
Residual	14.58	3	4.86			
Lack of fit	11.43	1	11.43	7.24	0.1148	Not significant
Pure error	3.16	2	1.58			
Cor total	674.15	10				
Std. dev.	2.20		R^2^		0.9784	
Mean	86.45		Adjusted R^2^		0.9279	

**Table 6 membranes-12-00420-t006:** ANOVA results for ammonia removal response.

Source	Sum of Squares	df	Mean Square	F-Value	*p*-Value	
Model	315.97	7	45.14	20.57	0.0154	Significant
A-DO	94.80	1	94.80	43.21	0.0072	
B-HRT	167.20	1	167.20	76.21	0.0032	
C-SRT	19.67	1	19.67	8.97	0.0579	
AB	16.97	1	16.97	7.74	0.0689	
AC	3.33	1	3.33	1.52	0.3059	
BC	12.70	1	12.70	5.79	0.0953	
ABC	1.29	1	1.29	0.5872	0.4993	
Residual	6.58	3	2.19			
Lack of fit	4.00	1	4.00	3.10	0.2206	Not significant
Pure error	2.58	2	1.29			
Cor total	322.55	10				
Std. dev.	1.48		R^2^		0.9796	
Mean	90.55		Adjusted R^2^		0.9320	

**Table 7 membranes-12-00420-t007:** ANOVA results for phosphorus removal response.

Source	Sum of Squares	df	Mean Square	F-Value	*p*-Value	
Model	121.55	7	17.36	12.77	0.0303	Significant
A-DO	81.66	1	81.66	60.06	0.0045	
B-HRT	28.67	1	28.67	21.09	0.0194	
C-SRT	3.05	1	3.05	2.24	0.2312	
AB	6.87	1	6.87	5.05	0.1101	
AC	0.4812	1	0.4812	0.3539	0.5938	
BC	0.1661	1	0.1661	0.1222	0.7498	
ABC	0.6548	1	0.6548	0.4816	0.5376	
Residual	4.08	3	1.36			
Lack of fit	0.0131	1	0.0131	0.0065	0.9433	Not significant
Pure error	4.07	2	2.03			
Cor total	125.63	10				
Std. dev.	1.17		R^2^		0.9675	
Mean	−10.15		Adjusted R^2^		0.8918	

## Data Availability

The data presented in this study are available on request from the corresponding author.

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
