# Peer review of "Effect of Operational Parameters on the Removal of Carbamazepine and Nutrients in a Submerged Ceramic Membrane Bioreactor"

_membranes, 2022, doi:10.3390/membranes12040420_

Round 1

Reviewer 1 Report

To Membranes

Title: Effect of operational parameters on removing carbamazepine and nutrients in submerged ceramic membrane bioreactor

Ref.: membranes-1657924

Original Paper

The Dao et al. reports on the removal of carbamazepine by submerged ceramic membrane bioreactor. In this study, the author used synthetic simulated wastewater and a bioreactor to investigate the removal rate of carbamazepine. Authors have studied hydraulic retention time, dissolved oxygen, and sludge retention time on removal. In addition, chemical oxygen demand, ammonia nitrogen, and phosphorus removal rate using ceramic membrane were investigated. The topic is relevant and important to journal criteria. The manuscript can be considered in Membranes journal after considering bellow mentioned minor comments.

  1. Abstract- Too many abbreviations, kindly remove all and keep MBR only as an abbreviation, others keep in introduction section.
  2. Paragraph 1 lines 37-42: There is no relevance to understanding terminology. The author has to provide literature.
  3. Figure 2 (a,b,c, and d) all needed error bar.

Reviewer 2 Report

The manuscript entitled “Effect of operational parameters on removing carbamazepine and nutrients in submerged ceramic membrane bioreactor” is dedicated to development of acceptable parameters for the use of ceramic membranes for pretreatment of wastewater discharged from medical institutions. The study touches on a very popular solution to reduce the release of medicinal metabolites into the environment. The authors carried out practical work when creating a cleaning problem as close as possible to the real one. When measuring HRT, DO, CZB, COD, NH4+-N, PO43--P, the validity of assumptions was analyzed using multifactor design. The results of the study demonstrated the ability to remove 38.36% CBZ with a ceramic membrane filter. Generally, the Manuscript is original and has a significance for the scientific community. Obtained results are reliable and supported by the data collected. The manuscript is easy to read and the arguments are described in a logical and understandable way.

In order to improve the manuscript, the following suggestion should be taken into account by the authors

Major issues:

  1. The amount of phosphates in the system of the proposed purification module is of concern. The increase in phosphorus content in the anaerobic-aerobic system is unclear. Explain how long the anaerobic regime was used and how long the aerobic one was carried out?
  2. On average, the dissolved oxygen (DO) content should be maintained at 1-1.5 mg/L in all parts of the aeration reactor. In 2.3. the chapter states that the dissolved oxygen content should be 2 mg/L, this meets the requirements of nitrification. NH4+-N under the action of oxygen in the air in such a structure is oxidized to nitrites, and then to NO3--N. NO3--Ncontains chemically bound O2 in its composition, which will lead to the oxidation of the organic substrate, which is so necessary for the normal course of dephosphotation. As a result, the question arises as to whether the chosen reactor mode does not contradict the conditions for improving the bio-removal process of PO43--P?
  3. If the purification method is used, without creating optimal conditions for biodegradation, the residual presence of phosphate compounds must be removed at the final stage at filtration facilities. As it is clear from the proposed conditions, the filter is an Al2O3 ceramic membrane. It is known that this membrane does not cope well with the removal of phosphates at low pH values. In this connection, the question arises about the pH measurements and their values in the experiment? Specify them.

Minor issues:

  1. The authors in Chapter 3.6 indicate that the high concentration of MLS in the reactor leads to starvation and cell death, and then releasing phosphorus. In this regard, the question arises whether the selected conditions lead to the degradation of carbamazepine, and not to its adsorption? Have membrane flushes been chemically tested, if not, why were they not considered useful?
  2. Please specify of activated sludges the minimum nutrients requisite for aerobic-growth based on BOD:N:P. (insert 2.3. section)

Author Response

Dear Reviewer, 

(Please see the attachment)

Thanks for the comments of the reviewer. However, it should be said that our research is aimed at optimizing the operating conditions for the best processing efficiency. We do not focus on the mechanism of removing pollutants, so it will be difficult for us to satisfy your questions as we do not have the necessary experimental data.

We hope you can consider and find our manuscript suitable for publication.

Sincerely,

Reviewer 3 Report

The manuscript investigate the effects of operating parameters (HRT, DO, and SRT) on removing carbamazepine, COD, ammonia nitrogen, and phosphorus using ceramic membrane bioreactor. Applying a two-level full-factorial design (FFD) analysis, optimum DO, HRT, and SRT 20 are obtained. The results show the applicability of the MBR system to treat wastewater with a high CBZ loading rate and the removal of nutrients. It is of interest to the scientists working in the field of MBR. The manuscript is clear and the experimental procedures described are in sufficient detail. This could be accepted with a minor reversion.

  1. In Abstract, Line14, Membrane technology (MBR) should be membrane bioreactor (MBR).
  2. The results of long-term operation under optimum parameters should be reported.
  3. How about the effect of membrane pore size on the removal rate?
  4. The manuscript requires minor spell check.

Reviewer 4 Report

I recommend the publication after minor corrections. Detailed comments are presented below.

Chapter 2.3.: The figure should not start the chapter, text with reference before the figure.

Line 114: explain the meaning of MLSS and MLVSS.

Figure 4 and 5: efficiency scale up to 100%.

Table 7 and Figure 6: text with reference before table and figure.

Line 305-307: could this phenomenon reduce the efficiency of removing other pollutants? Why was there a shortage of components, what was the load of pollutants on the sludge?

Line 332-338: I propose to present the application in a more general and understandable form. It may break it down into several separate conclusions about the factors affecting the removal of individual pollutants.
